# Dual Role of *Bacillus velezensis* EM-A8 in Maize: Biocontrol of Exserohilum Turcicum and Enhancement of Plant Growth

**DOI:** 10.3390/plants14223464

**Published:** 2025-11-13

**Authors:** María Fiamma Grossi Vanacore, Melina Sartori, Francisco Giordanino, Germán Barros, Daiana García

**Affiliations:** 1Laboratorio de Ecología Microbiana, Departamento de Microbiología e Inmunología, Facultad de Ciencias Exactas, Físico-Químicas y Naturales, Universidad Nacional de Río Cuarto, Ruta 36 km 601, Río Cuarto X5806JRA, Córdoba, Argentina; mgrossi@exa.unrc.edu.ar (M.F.G.V.); giordaninof@gmail.com (F.G.); 2Instituto de Ciencias de la Tierra, Biodiversidad y Ambiente, Consejo Nacional de Investigaciones Científicas y Técnicas, CONICET, Universidad Nacional de Río Cuarto, Ruta 36 km 601, Río Cuarto X5806JRA, Córdoba, Argentina; msartori@exa.unrc.edu.ar (M.S.); gbarros@exa.unrc.edu.ar (G.B.)

**Keywords:** maize, biocontrol agents, *Bacillus velezensis*, phytohormones, maize growth stimulation

## Abstract

Northern corn leaf blight (NCLB), caused by *Exserohilum turcicum*, is a major foliar disease of maize worldwide. To develop sustainable alternatives that reduce chemical products, we evaluated *Bacillus velezensis* EM-A8 (GenBank accession number OL704805) as a biocontrol agent under greenhouse and field conditions. The aims of this study were as follows: (i) characterize phytohormone production in two formulations containing the BCA; (ii) assess the influence of the BCA on plant biomass and yield; (iii) compare the efficacy of both formulations in controlling NCLB under field conditions; and (iv) determine whether the treatments affected salicylic acid and phenolic compound levels in maize tissues. The strain synthesized a broad spectrum of phytohormones, including salicylic acid, indoleacetic acid, indolebutyric acid, jasmonic acid, abscisic acid and gibberellic acid, as well as cytokinins such as kinetin, zeatin, and 6-benzylaminopurine. Foliar application increased maize dry biomass by 30%. In field trials, both formulations effectively suppressed NCLB, reducing the number of symptomatic leaves by 25–50% compared with controls. Furthermore, treated plants exhibited yield increases exceeding 1000 kg/ha. These findings demonstrate that *B. velezensis* EM-A8 provides effective biocontrol of *E. turcicum* while simultaneously enhancing maize growth and yield under field conditions. Future work should aim to scale up the use of *B. velezensis* EM-A8 in integrated pest management programs and evaluate its long-term impact on soil microbiota, plant health, and yield sustainability.

## 1. Introduction

Northern Corn Leaf Blight (NCLB) is one of the most widespread foliar diseases of maize, caused by *Exserohilum turcicum*. Global yield losses caused by NCLB are estimated at approximately 2.5% [1], corresponding to nearly 30 million tons of maize in 2023 [2]. In Argentina, where maize is one of the most important crops, NCLB causes significant yield reductions. For the 2024/2025 growing season, national maize production was around 50 million tons, and the planted area for the following season is estimated at 9.7 million hectares [3]. Over recent decades, the prevalence of NCLB in Argentinian maize has risen markedly, reaching nearly 90% annually [4]. In cases involving susceptible hybrids and favourable environmental conditions, yield losses can reach up to 40% [5]. The causal agent is a hemibiotrophic fungus that survives between growing seasons in crop residues and secondary hosts such as *Sorghum halepense* [6,7].

Management of this foliar disease primarily aims to avoid environmental conditions favorable for pathogen infection during the crop’s critical growth stages. In Argentina, however, late sowing dates are often preferred because they coincide with more stable rainfall patterns [8]. Another widely adopted strategy is chemical control through fungicide applications [9]. Although effective, this approach relies on moderately hazardous compounds and must be implemented before the disease exceeds a certain damage threshold [10,11]. In this context, biological control represents a promising alternative that can be integrated into pest management strategies [12]. Foliar pathogens interact with other phyllosphere microorganism, and the nature of these interactions—ranging from competition to antagonism—can influence crop health by inducing resistance to pathogens [13]. Previous studies conducted in our laboratory screened native maize phyllosphere microorganisms from their potential antagonistic activity [14,15,16]. In these studies, selected bacterial strains were evaluated for their antagonistic effects against *E. turcicum* both in vitro and in the field, including competition for nutrients, antibiosis, and effects on pathogen growth. Among the candidates, the biocontrol agent (BCA) *Bacillus velezensis* EM-A8 (GenBank accession number OL704805) was identified as the most promising, owing to its strong antagonistic properties and its tolerance to phyllosphere environmental conditions, as demonstrated by our research group [14,15,16].

In addition to suppressing pathogen, biocontrol agents (BCAs) can also promote plant growth. Several studies have demonstrated that certain *Bacillus* species not only control pathogens but also enhance maize growth under field conditions [17,18]. Both pathogen colonization and BCA activity are known to influence salicylic acid (SA) and phenolic compound levels—molecules central to physiological processes such as germination, flowering, and senescence, as well as plant responses to biotic and abiotic stresses [19,20,21,22]. Salicylic acid is a phenolic compound that promotes the expression of pathogenesis-related proteins that reinforce cell walls, induce lysis of infected cells, and trigger the hypersensitive response, ultimately leading to localized cell death [15,16]. Furthermore, it is a key molecule in the establishment of systemic acquired resistance (SAR) [23]. The response elicited by enhances plant defenses against subsequent infections, operating both locally and systemically [24]. While SA is strongly associated with resistance to biotrophic pathogens, its role in defence against necrotrophic and hemibiotrophic organisms remains less clearly defined [22].

Given the increasing prevalence of NCLB and the limitations associated with chemical control, the development of environmentally sustainable strategies for disease management has become a priority. Biological control using native microorganisms offers a promising approach to reduce pathogen pressure while minimizing the environmental impact of agriculture. In this context, *Bacillus velezensis* EM-A8, a strain previously isolated from the maize phyllosphere, represents a valuable alternative due to its antagonistic activity against *E. turcicum* and its potential to enhance plant growth. Evaluating the biocontrol efficacy and growth-promoting effects of this strain under field conditions is therefore of practical importance for improving maize productivity and sustainability in Argentina and other maize-producing regions. Due to this, the present study pursued the following objectives: (i) to evaluate the biostimulant activity of *B. velezensis* EM-A8 using two formulations under greenhouse and field conditions; (ii) to analyze its capacity for phytohormone synthesis in both formulations; (iii) to assess its biocontrol efficacy against NCLB under natural disease pressure; and (iv) to quantify total phenolic compounds in maize plants following foliar application of the BCA under field conditions with natural *E. turcicum* incidence.

## 2. Materials and Methods

### 2.1. Inoculum Preparation

*Bacillus velezensis* EM-A8 was originally isolated from the maize phyllosphere in corn fields located in the southern region of Córdoba Province, Argentina. Prior to this study, *B. velezensis* EM-A8 had been reported to exhibit protease and β-1,3-glucanase activities [25]. In addition, the strain was capable to producing several volatile organic compounds, including 2(3H)-Furanone, dihydro-3,5-Dimethyl; Silanediol dimethyl; Heptene,2,2,6,6-tetramethyl-4-methylene; 1-Pentene,2,4,4-trimethyl and Cyclopentasiloxane, decamethyl. Moreover, it displayed positive antibiosis activity against *E. turcicum*.

For this study work, two inoculum formulations were prepared: Formulation 1 (F1) consisted of cultures grown in nutrient broth, while formulation 2 (F2) was also supplemented with 5 g L^−1^ molasses and 10 g L^−1^ yeast extract [26]. Both formulations were adjusted with glycerol to reach 0.97 a_w_ and incubated for 24 h at 140 rpm and 25 °C until reaching final level of 10^8^ CFU mL^−1^.

### 2.2. Biostimulant Activity of B. velezensis EM-A8 Under Greenhouse Conditions

To evaluate the biostimulant effect of *B. velezensis* EM-A8 on maize, a greenhouse experiment was conducted using a randomized complete block design with three blocks. Each block consisted of ten replicates of maize plants cultivated in pots containing 0.5 L of natural soil. Treatments included foliar application of F1 and F2 at the V2 stage, with evaluations performed at the V4 stage (BBCH 10–19). Control plants (C) received no formulation. The variables measured were plant height, fresh and dry biomass. For dry biomass determination, plant material was placed in paper envelopes and oven-dried at 60 °C in a forced-air stove for 12 h. Fresh and dry biomass were quantified using a gravimetric scale. Data were subjected to analysis of variance (ANOVA), and means were compared using the DGC test [27] at a significance level of *p* < 0.05. Statistical analyses were performed using Infostat software InfoStat/L V. 2020. [28].

### 2.3. Phytohormone Determination and Quantification in Bacterial Formulations

To explore the potential mechanism underlying the biostimulant activity of *B. velezensis* EM-A8, the presence and concentration of phytohormone were determined in both F1 and F2. For extraction, 10 mL of each formulation was centrifuged, and the resulting supernatant was mixed with 2.5 mL of ethyl acetate. This mixture was centrifuged again, and the organic phase was recovered and evaporated to dryness. Samples were then resuspended in 100 µL of methanol, and 10 µL aliquots were injected into an Alliance 2965 chromatograph (Waters Inc., Boston, MA, USA, EE.UU.) Chromatographic separation was performed under a methanol: 0.2% acetic acid (40:60) gradient at a flow rate of 0.2 mL min^−1^. Data acquired and analysed were conducted using MassLynxTM 2.1 software with a Quatro UltimaTM Pt mass spectrometer (Micromass, Manchester, UK). Data acquisition and analysis were conducted using MassLynx™ 2.1 software with a Quattro Ultima™ Pt mass spectrometer (Micromass, Manchester, UK). Phytohormone quantification was based on calibration curves generated with known concentrations of each hormone and their corresponding deuterated internal standards. The HPLC-MS analysis was performed by the Plant Physiology Group (Grupo de Fisiología Vegetal, FCEFQyN, Universidad Nacional de Río Cuarto, Argentina). The phytohormones quantified included salicylic acid, indoleacetic acid, indolebutyric acid, jasmonic acid, kinetin, zeatin, benzylaminopurine, abscisic acid, and gibberellins.

### 2.4. Field Trials

The field trials to evaluated the biocontrol efficacy of *B. velezensis* EM-A8 formulations (F1 and F2) against *E. turcicum* were conducted in three maize fields located in the southern region of Córdoba, Argentina (coordinates: −33.013683, −64.132552; −33.009594, −64.111725; −33.006628, −64.488574). The trials were carried out during two consecutive maize growing seasons (2023–2025). The experimental design consisted of eleven randomized complete blocks, each comprising three treatments: foliar application of F1, foliar application of F2, and plants without inoculum control (C). Each experimental unit consisted of two rows, 20 m in length. Treatments were applied at the V8 phenological stage by foliar spraying (BBCH 10–19). Samples were randomly collected every 10 days from both treated and control plants. The following parameters were evaluated under field conditions after the *B. velezensis* EM-A8 application:

#### 2.4.1. NCLB Severity and Number of Affected Leaves

Total number of affected leaves per plant was recorded, and disease severity was assessed every ten days on the ear leaf and the leaves immediately above and below. Disease severity was measured according to the Bleicher scale (1988). Leaf tissue samples were also collected to evaluate phenolic compounds concentration. For this, samples were frozen at −80 °C until lyophilisation.

#### 2.4.2. Leaf Phenolic Compound Level

Total phenolic compounds were quantified from 0.1 g of lyophilized leaf tissue incubated for 2 h with 3 mL of methanol:water:hydrochloric acid (80:19:1). Samples were then centrifuged for 5 min at 3000 rpm, and 0.1 mL of supernatant was mixed with 0.75 mL of Folin–Ciocalteu reagent (10%). After 5 min, 0.75 mL of Na_2_CO_3_ was added. Absorbance was read at 725 nm after 90 min of reaction. The standard curve was prepared with different concentrations of gallic acid, ranging from 0.5 to 500 mg L^−1^. Results were expressed as mg gallic acid equivalent per g of dry weight.

#### 2.4.3. Yield Components

Yield components were estimated by determining (i) number of rows per ear; (ii) number of kernels per row, and (iii) weight of 1000 kernels. Measurements were taken from physiologically mature ears collected from 15 randomly picked plants per treatment and control. Kernel moisture content was measured with a hygrometer, and yield values were adjusted to 14.5% moisture.

### 2.5. Statistical Analysis

Analysis of variance (ANOVA) was performed for NCLB severity, number of affected leaves and phenolic compound concentrations, and yield components using Infostat software [29]. Means were compared using the DGC test at a significance level of *p* < 0.05.

## 3. Results

### 3.1. Biostimulant Activity of B. velezensis EM-A8 in the Greenhouse Trial

As described in materials and methods, plant height and biomass (fresh and dry) were evaluated to determine the biostimulant activity of *B. velezensis* EM-A8. Analysis of variance revealed that treatments had no significant effect on plant height (F: 1.55; d.f.: 2; *p* = 0.2185). Control plants reached an average height of 47.23 cm, while plants treated with F1 and F2 had 46 and 49.51 cm, respectively. Fresh biomass was higher in treatment F2 (3.77 g plant^−1^) compared to F1 (3.15 g plant^−1^) and the control (3.13 g plant^−1^), although these differences were not statistically significant (F: 2.13; d.f.: 2; *p* = 0.1255). In contrast, dry biomass production showed significant differences among treatments (F: 3.68; d.f.: 2; *p* = 0.0296). Plants treated with F2 produced significantly more dry biomass (0.44 g plant^−1^) than those treated with F1 (0.37 g plant^−1^) or the control (0.34 g plant^−1^) (Figure 1).

### 3.2. Phytohormones Detection and Quantification in Formulations

Phytohormone profiling of the bacterial formulations demonstrated that *B. velezensis* EM-A8 is capable of synthesizing growth-promoting compounds. As shown in Table 1, five out of the nine phytohormones assessed were detected in F1, including salicylic acid, indoleacetic acid, kinetin, zeatin, and 6-benzylaminopurine. In contrast, F2 contained all nine phytohormones evaluated and at higher concentrations than those found in F1.

### 3.3. Field Trial

#### 3.3.1. NCLB Severity and Number of Affected Leaves

NCLB severity and number of affected leaves were evaluated across three fields at different sampling times. In Field 1, which was evaluated in the same campaign year as Field 2, no significant differences among treatments were detected for severity (F = 1.17; d.f. = 2; *p* = 0.3167) or number of affected leaves (F = 1.58; d.f. = 2; *p* = 0.2117). Mean values were 5.59 ± 3.46% for severity and 2.62 ± 0.70 for the number of affected leaves (Figure 2).

In Field 2, ANOVA revealed significant differences in NCLB severity (F = 4.14; d.f.: 2; *p* = 0.0187). Control plants showed significantly higher severity (3.78%) compared to F1 (3.06%) and F2 (3.19%). However, the number of affected leaves was not significantly different among treatments (F = 2.96; d.f. = 2; *p* = 0.0561), with an overall mean of 2.59 ± 0.76 affected leaves.

In Field 3, conducted two years later, no significant differences were observed among treatments for NCLB severity (F = 1.36; gl = 2; *p* = 0.2615), with mean values of 1.67 ± 1.42%. Similarly, the number of affected leaves was not significantly different (F = 2.56; gl = 2; *p* = 0.0819) averaging 1.34 ± 0.95 leaves. However, when analysing individual sampling times, significant differences were detected 47 days after treatment application (sampling time 4) (F = 9.50; d.f. = 2; *p* = 0.0034). At this point, control plants had more affected leaves (3.67) compared to F1 (2.50) and F2 (2.33).

#### 3.3.2. Leaf Phenolic Compound Concentration

ANOVA revealed significant differences in phenolic compound concentration between fields (F = 6.3; gl = 2; *p* = 0.0025). Field 3 presented the highest levels (18.96 mg GAeq per g of dry weight) followed by fields 2 (16.06 mg GAeq per g of dry weight) and Field 1 (15.61 mg GAeq per g of dry weight) (Figure 3). However, within each field, treatments did not significantly affect phenolic compound concentrations (Field 1: F = 0.05; d.f. = 2; *p* = 0.9473; Field 2: F = 1.52; d.f. = 2; *p* = 0.2465; Field 3: F = 0.39; d.f. = 2; *p* = 0.6801).

#### 3.3.3. Yield Components

Yield component data are shown in Table 2. Respect to the number of rows per ear, was not affected by the application of the treatments for fields 1 and 3. In Field 2, however, control showed significantly higher values (*p* = 0.0304) than F1 and F2.

Regarding the number of kernels per row, in field 1, no significant differences were found, with mean values ranging from 27.67 to 28.27. In Field 2, F2 significantly increased kernel number compared to F1 and C (*p* = 0.0031). In Field 3, both formulations (F1 and F2) resulted in significantly higher values compared to the control (*p* < 0.0001)

Thousand kernel weight (g) was significantly affected by treatments in all fields (*p* < 0.0001). In Field 1, the highest weight was obtained for F2 treatment, followed by F1 and C. In Field 2, F1 showed the highest value followed by F2 and C. In contrast, in Field 3 this parameter was significantly higher in control (*p* < 0.0001), followed by F1 and F2.

The estimated yield was significantly higher in C and F2 in Field 1 (*p* < 0.0001), but C yielded significantly less in Field 2 (*p* < 0.0001) obtaining significantly higher values for the treatments. There were no statistical differences between treatments for grain yield in Field 3 (*p* = 0.4628).

## 4. Discussion

In this study, we evaluated a biological control strategy for northern corn leaf blight (NCLB). This disease is typically managed through early sowing or chemical fungicides applications; however, these measures are often insufficient to provide reliable protection. Moreover, synthetic fungicides can be hazardous to human and animal health, leaving toxic residues in the environment. In this sense, biological control is an alternative strategy to the use of chemical compounds, involving the use of beneficial microorganisms for disease control. Biological control represents a sustainable alternative, relying on beneficial microorganisms to suppress plant diseases. In this context, we assessed both the biocontrol and biostimulant activities of *Bacillus velezensis* EM-A8 in maize.

### 4.1. Biostimulant Activity of B. velezensis EM-A8 Under Greenhouse Condition

As a first step, a greenhouse trial was carried out to determine whether the laboratory-prepared formulations (F1 and F2) exerted a biostimulant effect on maize growth. The results of this assay demonstrate that the bacteria promote plant growth, as evidenced by a 30% increase in dry biomass production. One of the mechanisms by which growth-promoting bacteria exert their biostimulant effect is by through the production of phytohormones that regulate plant development [29,30,31]. Therefore, we analysed whether B. velezensis EM-A8 releases phytohormones related to growth and defence. The results revealed the presence of five phytohormones in F1 and nine in F2. Moreover, the phytohormones common to both formulations were detected at higher levels in F2. This difference in the type and concentration may be attributable to the composition of the liquid media used for formulation production. In the case of F2, the medium was enriched with molasses and yeast extract, which may have stimulated metabolite production. In both formulations, salicylic acid was the predominant phytohormone. This compound plays a central role in vital processes such as stress tolerance, activation of defence mechanisms, photosynthesis and growth and development [20,21,22]. In maize, [30] and Ali et al. [32] reported that salicylic acid alleviates the deleterious effects of salinity and enhances grain yield, while Li et al. [33] demonstrated that exogenous salicylic acid maintains photosynthetic rates under heat stress. With respect to defence, salicylic acid enhances antioxidant enzyme activity, regulates cell wall strengthening, and induces systemic acquired resistance [34]. Consistent with this, Li et al. [35] observed that a maize mutant with enhanced resistance to *Curvularia lunata* accumulated higher levels of salicylic acid and jasmonic acid.

Many studies have demonstrated that certain strains of fungi and bacteria can stimulate plant growth through different mechanisms [36,37,38,39,40,41]. For example, Adesemoye et al. [36], evaluated *Bacillus subtilis* and *Pseudomonas aeruginosa* strains in *Solanum lycopersicum* L. (tomato), *Abelmoschus esculentus* (okra), and *Amaranthus* sp. reported up to 30% increase in dry biomass after 60 days, for both bacterial strains. While many studies have documented plant growth stimulation by applying microorganisms to the rhizosphere or seeds, to our knowledge, no studies have evaluated microbial application directly to the foliar area, as proposed in this research.

### 4.2. Determination of Phytohormone Profiles in Formulations

Indoleacetic acid (IAA) and indolebutyric acid (IBA) are auxins naturally present in plants that regulate key processes of growth and development, including root elongation, shoot formation, leaf morphogenesis, and kernel development. Regarding grain yield, IAA has been shown to stimulate sugar and protein metabolism during kernel differentiation and to promote nutrient allocation to the endosperm [42,43]. In this context, IAA production by plant growth-promoting Bacillus species enhances root and coleoptile elongation [44]. Beyond their developmental roles, auxins also participate in plant defense mechanisms. Our phytohormone analysis of formulations F1 and F2 confirmed that *B. velezensis* EM-A8 produces both IAA and IBA, which may partially explain the increased dry biomass and yield observed under greenhouse and field conditions. Consistently, Shi et al. [45] reported that 45 auxin-related genes were differentially expressed during the defensive response of maize to Setosphaeria turcica, suggesting that auxin signaling contributes to the modulation of plant immunity against this pathogen.

The detection of jasmonic acid in F2, suggests that *B. velezensis* EM-A8 may contribute to the activation of defense signaling pathways in maize. Jasmonic acid is a key phytohormone involved in resistance to necrotrophic pathogens and in the activation of induced systemic resistance (ISR). In agreement with our findings, Lipps et al. [46] reported that jasmonic acid biosynthetic pathways are associated with enhanced resistance to maize leaf blight, while Liu et al. [46,47] demonstrated that jasmonic acid biosynthesis is rapidly induced -within the first 72 h after infection by *S. turcica*- and plays a crucial role in the plant’s defensive response. These results support the hypothesis that jasmonic acid introduction by *B. velezensis* EM-A8 in F2 could be an important mechanism underlying its biocontrol activity against NCLB.

Cytokinins (CK) are phytohormones that induce cell division and regulate multiple developmental processes, including nutrient mobilisation, seed germination, root growth, stress response, and apical dominance [48]. In both formulation (F1 and F2), the presence of the cytokinins kinetin, zeatin and 6-benzylaminopurine was detected, which may help explain the higher biomass observed in the greenhouse trial and the increased grain yield recorded in the field trial. Previous studies have reported that certain biocontrol agents (BCAs) can produce CKs and enhance plant growth and defense. For instance, Bean et al. [49] demonstrated that *Trichoderma* strains produce CKs and positively influence the resistance of *Arabidopsis* spp. to *Fusarium graminearum*. However, to date, there is no evidence of a role for CKs in the biological control of Northern Corn Leaf Blight (NCLB).

Abscisic acid is involved in plant development as well as in abiotic stress resistance. This phytohormone mitigates the harmful effects of drought, chilling and soil salinity, thereby contributing to yield improvement [38,45]. In this research, abscisic acid was detected in F2 but not in F1, which could partially account for the differences in dry biomass production observed between the two formulations of the same bacterium. Balmer et al. [50] suggested that ABA participates as a chemical regulator of root-to-leaf systemic acquired resistance (SAR) in maize, and proved that ABA treatment of roots reduced the fungal growth of *C. graminicola* on leaves, thus mimicking biological SAR. These results are consistent with Erb et al. [51], who provided evidence that ABA is involved in defence gene induction and resistance to the *S. turcica* pathogen.

Gibberellins are ubiquitous phytohormones that regulate plant growth by promoting cell division, stem elongation and grain development and mitigate abiotic stresses [52,53,54]. In maize, Cui et al. [55] reported that gibberellin application at two phenological stages increased grain yield through different yield components, such as grain weight or kernel number, depending on the timing of application. In this work, gibberellins were detected in F2 but not in F1. Their presence in F2 may help explain the greater biomass accumulation observed in plants treated with this formulation in the greenhouse trial.

Beneficial bacteria provide a wide range of plant growth-promoting abilities for agricultural use, such as biocontrol activity, phytohormone production and nutrient solubilisation. However, the effectiveness of these bacteria under field conditions may differ from greenhouse or laboratory results due to environmental variations [56]. Therefore, effective integration of such tools into production systems requires scaling the experiments to field conditions to confirm bacterial functionality. Since our greenhouse trial demonstrated the biostimulating activity of *B. velezensis* EM-A8 and Sartori et al. [15,16] confirmed its ability to control NCLB, we conducted a field trial to assess its biocontrol capacity, its effect on foliar concentration of phenolic compounds, and its impact on grain yield in maize under field conditions.

### 4.3. NCLB Severity and Number of Affected Leaves

The three field trials revealed a decrease in NCLB severity and in the number of affected leaves following the application of F1 and F2. These findings are in line with Sartori et al. [15], who reported lower incidence of NCLB and common rust in maize after treatment with two antagonists, one of which was *B. velezensis* EM-A8. Similar results have been described in the same pathosystem by Zhang et al. [57], who observed that *Klebsiella jilinsis* 2N3 reduced NCLB severity, and by Ding et al. [56], who showed inoculation with *B. subtilis* DZSY21 decreased NCLB intensity in maize. Likewise, Chen et al. [58] found that treatment with *Paenibacillus polymyxa* SF05 significantly reduced the disease index of banded leaf and sheath blight.

### 4.4. Leaf Phenolic Compound Concentration Level

Phenolic compounds constitute a broad group of metabolites that play an important role in resistance to pathogens including resistance to fungal disease [59]. In plant-beneficial microbe interaction, several *Bacillus* spp. have been shown to induce systemic resistance by enhancing phenolic compounds accumulation [60]. However, in our study, no significant differences in leaf phenolic compound concentration were observed between treatments and the control. Thus, we could not demonstrate that *B. velezensis* EM-A8 enhances maize resistance to NCLB through phenolic compounds accumulation under field conditions. In contrast, Wallis & Galarneau [61] reported that phenolic compounds generally increased in plants following interactions with beneficial microbes, pathogens and insects. Similarly, Li et al. [35] found elevated levels of phenylpropanoids in a maize mutant line resistant to *Curvularia lunata*.

### 4.5. Yield Components

Grain yield in the field trials was estimated based on its components: number of rows per ear, number of kernels per row and thousand-kernel weight. The number of rows per ear did not differ significantly among treatments, averaging around 16 rows. In contrast, application of F1 and F2 increased the number of kernels per row by 4.11% and 8.33%, respectively, in. This is consistent with Satorre et al. [62], who reported that the number of rows per ear is more strongly determined by genetics while the environment exerts less influence on this trait in the selected hybrid. Thousand-kernel weight was also significantly higher in F1 (9.22%) and F2 (11.29%) compared to the control. Because grain yield is directly affected by biomass production [11], the greenhouse trial results support the expectation of yield improvements not only due to reduced NCLB severity but also through the biostimulant activity of the bacteria. The superior performance of F2 relative to F1 may be explained by the presence of gibberellins, which are known to stimulate plant growth. In this sense, Balderas-Ruíz et al. [29] showed that *B. velezensis* promoted biomass accumulation in mango plants in addition to its biocontrol effect against anthracnose. Etesami et al. [30] reviewed multiple applications of *Bacillus* spp. and concluded that *B. velezensis* can enhance crop yields by producing phytohormone and other growth regulators that alleviate stress and promote plant growth.

### 4.6. Future Projections

Future research should focus on elucidating the molecular and physiological mechanisms underlying these biocontrol and biostimulant effects, evaluating the consistency of strain performance across different maize genotypes and agroecological environments, and optimizing formulation stability for large-scale field application.

## 5. Conclusions

In conclusion, both formulations of *B. velezensis* EM-A8 demonstrated efficacy in controlling NCBL under field conditions and exhibited biostimulant activity in maize. In greenhouse trials, increased biomass production was observed, while in field trials, yield improvements were recorded. These finding indicate that *B. velezensis* EM-A8 formulations have a dual potential, serving both as a preventive strategy against E. turcicum and as a biostimulant to enhance maize productivity.

## Figures and Tables

**Figure 1 plants-14-03464-f001:**
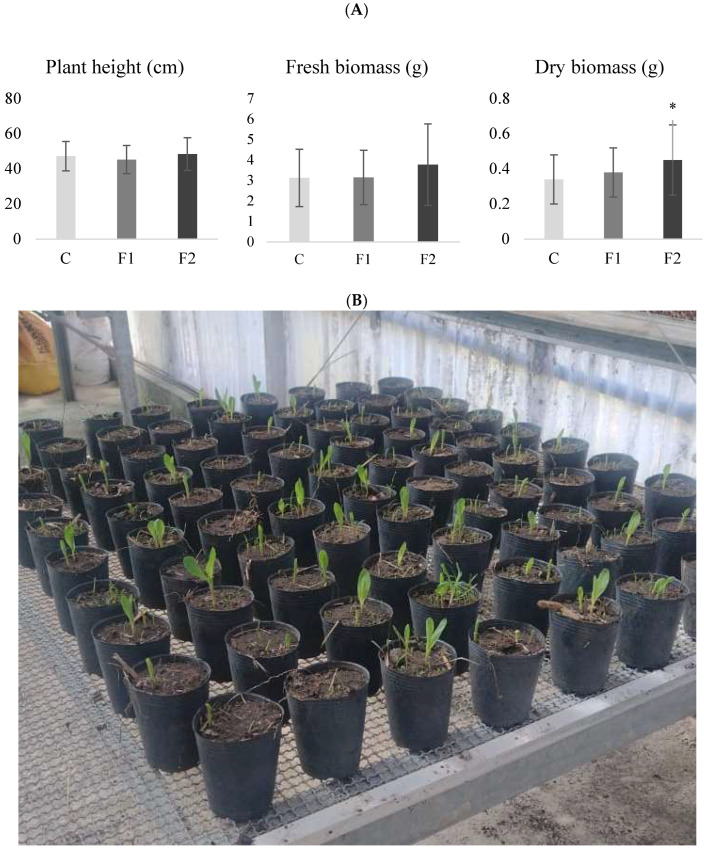
Maize growth parameters for each treatment. (**A**): biomass (g) means and standard deviations for formulate 1 (F1), formulate 2 (F2) and control (C) in the greenhouse trial. (**B**): Pots containing maize plants at the beginning of the biostimulant activity assay of the formulations. The asterisk (*) indicates significant differences for biomass dry weight according to the DGC test at a significance level of *p* < 0.05.

**Figure 2 plants-14-03464-f002:**
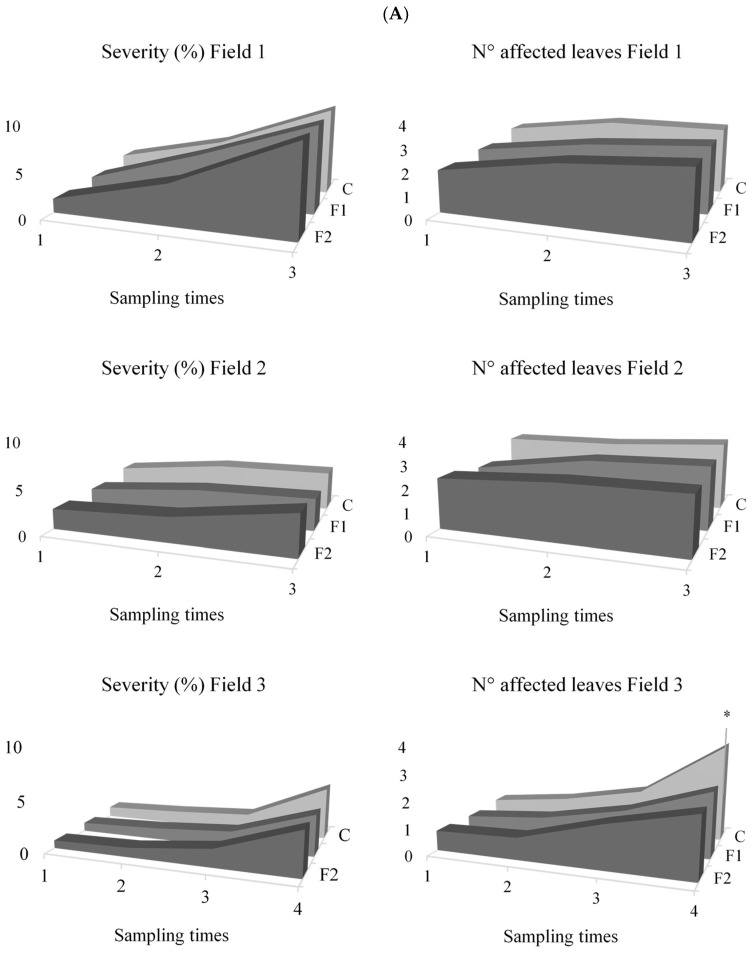
(**A**): Severity and incidence (number of affected leaves) of NCLB across sampling dates for each treatment (F1, F2, and Control) in Fields 1, 2, and 3. (**B**): Photographs of the corn trials, with stakes marking the furrows to separate the different treatments. The asterisk (*) indicates significant differences for N° of affected leaf in the Field 3 according to the DGC test at a significance level of *p* < 0.05.

**Figure 3 plants-14-03464-f003:**
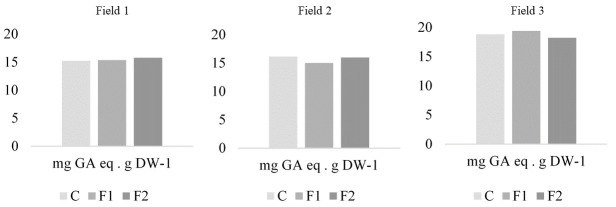
Phenolic compound concentration (mg GA eq g DW^−1^) for each treatment in the three fields, for formulate 1 (F1), formulate 2 (F2) and control (C).

**Table 1 plants-14-03464-t001:** Mean values and standard deviations of phytohormones detected in both formulations (F1 and F2) produced by *Bacillus velezensis* EM-A8. Results are expressed in ng·mL^−1^. SA: salicylic acid; IAA: indole-3-acetic acid; IBA: indole-3-butyric acid; JA: jasmonic acid; KIN: kinetin; Zea: zeatin; 6-BAP: 6-benzylaminopurine; ABA: abscisic acid; GA_3_: gibberellic acid; nd: not detectable.

Phytohormone	F1	F2
SA	1329.4 ± 70.01	2989.2 ± 368.57
IAA	52.4 ± 4.61	61 ± 2.00
IBA	nd	62.1 ± 8.78
JA	nd	12.3 ± 3.05
KIN	107.9 ± 28.42	106.6 ± 8.65
ZEA	105 ± 6.25	109.4 ± 15.43
6-BAP	101.6 ± 16.51	112.9 ± 25.95
ABA	nd	16.4 ± 6.21
GA-3	nd	24.6 ± 7.2

**Table 2 plants-14-03464-t002:** Means and standard deviation of yield parameters for treatments and control in Fields 1, 2 and 3, for formulate 1 (F1), formulate 2 (F2) and control (C). Means with (*) and (**) are statistically different.

		Field 1	Field 2	Field 3
N° rows per ear	C	16.44 ± 1.80	* 16.00 ± 1.52	15.20 ± 1.38
	F1	16.27 ± 1.68	15.30 ± 1.59	15.10 ± 1.53
	F2	16.27 ± 1.38	15.50 ± 1.30	15.60 ± 1.36
N° kernels per row	C	28.27 ± 3.87	26.97 ± 6.41	25.08 ± 4.20
	F1	27.67 ± 4.17	27.87 ± 5.38	* 27.65 ± 4.01
	F2	27.73 ± 3.31	* 30.41 ± 5.16	* 28.18 ± 3.34
Thousand kernel weight (g)	C	192.66 ± 2.38	196.18 ± 6.84	** 309.80 ± 14.16
	F1	* 200.967 ± 10.67	** 227.35 ± 48.76	* 293.93 ± 24.49
	F2	** 223.07 ± 13.95	* 208.76 ± 26.68	280.10 ± 21.46
Yield (kg ha^−1^)	C	* 8835.76 ± 1350.10	6434.58 ± 1385.71	9592.02 ± 2020.82
	F1	7856.53 ± 1717.52	* 8895.30 ± 3009.47	9955.80 ± 1978.57
	F2	* 9217.66 ± 1113.32	* 8332.53 ± 2093.78	9965.30 ± 1592.60

## Data Availability

We declare that all published data is that which appears in the article. We have no additional data.

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
