# Peer review of "Dual Role of *Bacillus velezensis* EM-A8 in Maize: Biocontrol of Exserohilum Turcicum and Enhancement of Plant Growth"

_plants, 2025, doi:10.3390/plants14223464_

Round 1
Reviewer 1 Report
Comments and Suggestions for Authors
In this manuscript, Maria Fiamma Grossi Vanacore and colleagues evaluated a biocontrol agent previously isolated and investigated its effects on maize under both greenhouse and field conditions.. I have following comments:
1, For the Abstract section, the present version is a bit too long, please modify.
2, For the introduction section, practical interests of this study be included.
3, Methods for random sampling and statistical analysis should be introduced in the Methods.
4, For Figure 1, growth picture of maize plants should be exhibited.
5, For Table 1, standard error value and results of significance different analysis should be exbited.
6, For Figure 2, picture showing maize plants in field should be exhibited.
7, For the discussion, I suggest that the discussion section should be divided into subsections with appropriate titles.
Author Response
Reviewer 1
Thank you very much for your careful review of our manuscript. We appreciate your insightful comments and suggestions, which we believe will significantly improve the quality of our publication. Below, we provide point-by-point responses to each of your comments.
1, For the Abstract section, the present version is a bit too long, please modify.
Done. The abstract was modified.
2, For the introduction section, practical interests of this study be included.
We appreciate the reviewer’s valuable suggestion. Initially, we assumed that this aspect was implicit; however, upon review, we recognized the importance of addressing it explicitly. Therefore, a new paragraph has been added on lines 89–98 and highlighted in yellow for clarity.
3, Methods for random sampling and statistical analysis should be introduced in the Methods.
The statistical methods used are described in Section 2.5 of the Methodology. If the reviewer requires further clarification, we can provide additional details. Regarding the random field sampling, a detailed diagram of the field trial and its corresponding sampling scheme can also be included upon request. In any case, the sampling information, which was not previously reported, has now been added (L-158-159)
4, For Figure 1, growth picture of maize plants should be exhibited.
Done, the figure with the assay in the greenhouse was added under figure 1. We added a photo from the first days of the trial; the pots with the plants are inside our greenhouse.
5, For Table 1, standard error value and results of significance different analysis should be exhibited.
Done.
6, For Figure 2, picture showing maize plants in field should be exhibited.
Four pictures in a collage were added as Figure 2 B.
7, For the discussion, I suggest that the discussion section should be divided into subsections with appropriate titles.
Done

Reviewer 2 Report
Comments and Suggestions for Authors
Overall, the manuscript appears interesting and well-written; however, it requires several revisions.
Abstract
Please provide information about future research.
Introduction
I suggest slightly shortening the part about NCLB
I recommend adding a paragraph about the novelty of the study.
Materials and Methods
Lines 114–115 – Please consider adding the growth stage according to the BBCH scale.
Line 123 – Please indicate whether previous studies investigated biocontrol factors such as antibiotic lipopeptides (e.g., surfactin, iturin, fengycin) or chitinases. If so, please include this information in section 2.1.
Line 167 – Please introduce subscripts and check the entire manuscript for similar cases.
Results
Please add tables and figures to the manuscript.
Line 194 – Please apply superscripts.
Discussion and Conclusion
I strongly recommend elaborating the discussion in terms of biocontrol, e.g., writing more about Jasmonic acid (JA) (line 297) in this context.
Lines 279, 284, 324, 347 – Please correct the citations according to MDPI requirements and check all references throughout the manuscript.
I suggest adding some information about future research if planned, e.g., on other pathogens or plants.
Author Response
Thank you very much for your careful review of our manuscript. We appreciate your insightful comments and suggestions, which we believe will significantly improve the quality of our publication. Below, we provide point-by-point responses to each of your comments.
Reviewer 2
Abstract
Please provide information about future research.
Done
Introduction
I suggest slightly shortening the part about NCLB
Done
I recommend adding a paragraph about the novelty of the study.
We appreciate the reviewer’s valuable suggestion. Initially, we assumed that this aspect was implicit; however, upon review, we recognized the importance of addressing it explicitly. Therefore, a new paragraph has been added on lines 89–98 and highlighted in yellow for clarity.
Materials and Methods
Lines 114–115 – Please consider adding the growth stage according to the BBCH scale.
Done. This was made in parenthesis and added also in the field assay.
Line 123 – Please indicate whether previous studies investigated biocontrol factors such as antibiotic lipopeptides (e.g., surfactin, iturin, fengycin) or chitinases. If so, please include this information in section 2.1.
Done, this information was provided in 2.1 section in L-100-106.
Line 167 – Please introduce subscripts and check the entire manuscript for similar cases.
Done
Results
Please add tables and figures to the manuscript.
Done, at the final of the manuscript we added the tables and figures
Line 194 – Please apply superscripts.
Done
Discussion and Conclusion
I strongly recommend elaborating the discussion in terms of biocontrol, e.g., writing more about Jasmonic acid (JA) (line 297) in this context.
Done. It was revised in all the discussion
Lines 279, 284, 324, 347 – Please correct the citations according to MDPI requirements and check all references throughout the manuscript.
Done
I suggest adding some information about future research if planned, e.g., on other pathogens or plants.
Done, it was added at the final of the “discussion section” in L-389-392

Round 2
Reviewer 1 Report
Comments and Suggestions for Authors
Authors have addessed my concerns in the revision.
Reviewer 2 Report
Comments and Suggestions for Authors
After revision, I recommend this manuscript for publication.